# Assessment of Toxigenic Fusarium Species and Their Mycotoxins in Brewing Barley Grains

**DOI:** 10.3390/toxins11010031

**Published:** 2019-01-10

**Authors:** Karim C. Piacentini, Liliana O. Rocha, Geovana D. Savi, Lorena Carnielli-Queiroz, Livia De Carvalho Fontes, Benedito Correa

**Affiliations:** 1Biotechnology Department, University of Sao Paulo, Av. Professor Lineu Prestes, Sao Paulo 2415, Brazil; correabe@usp.br; 2Department of Food Science, Food Engineering Faculty, University of Campinas, Av. Monteiro Lobato, 80, Campinas 13083862, Brazil; l.rocha@unicamp.br; 3Department of Materials Sciences and Engineering, University of the Extreme Southern Santa Catarina, Av. Universitaria, 1105 Criciuma, Santa Catarina 88807-400, Brazil; geovanasavi@gmail.com; 4Microbiology Department, University of Sao Paulo, Av. Professor Lineu Prestes, Sao Paulo 1374, Brazil; carnielli@usp.br (L.C.-Q.); livia.fontes@usp.br (L.D.C.F.)

**Keywords:** cereals, mycotoxigenic fungi, phylogeny, deoxynivalenol, zearalenone

## Abstract

*Fusarium* species threaten yield and quality of cereals worldwide due to their ability to produce mycotoxins and cause plant diseases. Trichothecenes and zearalenone are the most economically significant mycotoxins and are of particular concern in barley, maize and wheat. For this reason, the aim of this study was to characterize the *Fusarium* isolates from brewing barley and to assess deoxynivalenol and zearalenone contamination in grains. Characterization of the *Fusarium* strains was carried out by the phylogeny based on two loci (EF-1α and RPB2). Mycotoxin detection and quantification were performed by LC-MS. The results show that *Fusarium* was the predominant genus. Phylogenetic study demonstrated that the majority of the strains clustered within the *Fusarium sambucinum* species complex followed by the *Fusarium tricinctum* species complex. The results revealed high incidence of deoxynivalenol (DON) and zearalenone (ZEA) contamination (90.6% and 87.5%, respectively). It was observed that 86% of the samples contaminated with ZEA were above the limits set by the EU and Brazilian regulations. These results may highlight the importance of controlling *Fusarium* toxins in barley, mainly because of its use in the brewing industry and the resistance of various mycotoxins to food processing treatments.

## 1. Introduction

The *Fusarium* genus includes plant pathogens which are of great concern to agricultural production and food/feed safety worldwide [1], threatening yield and quality of cereals and producing mycotoxins, secondary metabolites that are toxic to humans and other animals [2]. *Fusarium* genus is capable of producing several mycotoxins, including fumonisins, trichothecenes and zearalenone. These are the most economically significant *Fusarium* mycotoxins and are of particular concern in barley, maize and wheat [3].

Trichothecenes have been classified into four groups: types A–D, according to their chemical structure [4], the most important in cereals are types A and B [1]. The B-trichothecenes include the mycotoxins deoxynivalenol (DON), its acetylated derivatives, 3-acetyldeoxynivalenol (3ADON) and 15-acetyldeoxynivalenol (15ADON) and nivalenol (NIV). DON is the most frequent type-B trichothecene and can be found worldwide. Furthermore, DON inhibits protein synthesis and it has been associated with intoxication of animals through consumption of contaminated feed [5].

Zearalenone (ZEA) is a mycotoxin with estrogenic effects produced by several *Fusarium* species and is usually found in cereal grains. Swine are especially sensitive to the estrogenic effects of ZEA. This toxin has been shown to bind to the estrogenic receptors and to inhibit ovulation. It is, therefore, often involved in hormonal disorders of farm animals and it is also related to hypoestrogenic syndromes in humans [6]. ZEA has been classified into group 3 (non-classifiable due to its carcinogenicity to humans) by IARC (International Agency for Research on Cancer) [7].

During recent years, mycotoxins have attracted international attention not only for their perceived impact on human health but also because of the economic losses accruing from contaminated foods. Considering barley matrix, several international studies have reported on *Fusarium* and its mycotoxins contamination due to the beer gaining increased popularity [8]. The main problem is the characteristics that these compounds have. Some mycotoxins such as trichothecenes, zearalenone and fumonisins are considered stable during the brewing process [9] and can persist to the final product, the beer. Furthermore, *Fusarium* infection causes a negative impact on barley germination rates which results in malting quality and yield reduction. Additionally, this produces gushing and changes in color and flavor of the beer [10].

Barley is considered to have good characteristics for fungi contamination under favorable conditions. In addition, environmental factors associated with barley varieties and agronomic practices influence the *Fusarium* infection and the mycotoxin production. The climate conditions during critical phases of barley plant growth could lead to noticeable difference between the mycotoxin diversity. Tropical conditions, such as those found in Brazil contribute to fungi dissemination and consequently mycotoxin production in barley. For example, 2015 demonstrated a high rainfall average and high levels of humidity, which prompted worse contamination levels found until now [11]. It is necessary to mention that only two *Fusarium* mycotoxins were analyzed. Other fungi mycotoxins should be evaluated along with their masked toxins.

All of the barley harvested in Brazil is destined for the brewing industry and the production continues to increase. Southern Brazil has the largest number of barley-producing regions, therefore, the quality of the grains needs to be monitored and studies about the fungi profile should be taken into account. The knowledge of the current contamination of barley in the region, as well as the constant monitoring, is necessary in order to evaluate if agronomic practices are being duly effective to control the contamination in grain production. The irrigation management, resistant cultivars, harvesting strategies, chemical and biological control and disease forecasting could minimize the loss of grain quality and avoid the disease caused by mycotoxigenic fungi [12].

The current occurrence of mycotoxins in barley could lead to the necessity of developing new strategies or improving those currently in place for more effective management of mycotoxins in the future. This fact is even more relevant when taking into account that the Brazilian regulation for DON in barley will be updated by 2019 [13] and therefore, the information on the occurrence of this toxin is still discussed in the country often.

The regulations have set maximum levels for mycotoxin contamination in grains, in order to avoid further accumulation of mycotoxins in processed food and to control mycotoxin of major concern in unprocessed cereals, such as brewing barley, with 1.250 µg/kg being the maximum permitted for DON and 100 µg/kg for ZEA. From January 2019, DON limits for brewing barley will be set at 1000 µg/kg [13]. Similarly, the limits for DON and ZEA fixed by the European Commission [14] are equal to 1.250 µg/kg and 100 µg/kg for unprocessed cereals.

For the reasons stated above, the aim of the present research was to characterize the *Fusarium* isolates and to assess DON and ZEA contamination in brewing barley grains. These findings provide new insights into the diversity of *Fusarium* species isolated from Brazilian barley and add information to the mycotoxin profile in a source (raw material) destined for the food industry.

## 2. Results

### 2.1. Water Activity and Identification of Isolated Fungi

Water activity levels ranged from 0.579 to 0.667 (mean: 0.622 ± 0.02) for all of the 64 brewing barley samples analyzed. Filamentous fungi were isolated from 96.8% of the samples, highlighting that *Fusarium* was the predominant genus (46%), followed by *Alternaria* (28.8%), *Phoma* (15%), *Epicocum* (6.2%), *Penicillium* (2%), *Aspergillus* (1.1%) and *Rhyzopus* (0.9%). Species belonging to *Fusarium* genus were initially identified through the sequencing of the *EF-1*α locus. Sequences were submitted to Blast (basic local alignment) tool on NCBI database (https://blast.ncbi.nlm.nih.gov/Blast.cgi?PAGE_TYPE=BlastSearch).

The identification analysis was carried out with 48 *Fusarium* strains and sequencing analysis determined 56.26% of the *Fusarium* species isolated in this study, were within the *Fusarium sambucinum* species complex (FSAMSC), 31.25% within the *Fusarium tricinctum* species complex (FTSC), 8.33% within the *Fusarium fujikuroi* species complex (FFSC) and 2% within both, *Fusarium incarnatum-equiseti* (FIESC) and *Fusarium oxysporum* species complexes (FOSC) (Table 1).

### 2.2. Phylogenetic Study

The phylogenetic study was conducted for the species complexes that have the potential to produce trichothecenes and for the closely related species complex *F. tricinctum.* The concatenated loci *EF-1*α and *RPB2* were used to infer the phylogeny of the species isolated from barley. The data set consisted of 65 taxa, 1292 nucleotides with 452 parsimony-informative characters (PICs) (Figure 1). The analysis resulted in a one most parsimonious tree (CI = 0.69 RI = 0.94). The majority of the isolates clustered within *F. graminearum*, *F. poae* and *F. avenaceum* species complexes with both posterior probability and bootstrap supports (Figure 1).

The method validation and recovery experiments are summarized in Table 2 and Appendix A. The reported LODs were set at 5 and 10 µg/kg for DON and ZEA, respectively and LOQs 25 µg/kg for both toxins. The coefficients of correlation (R^2^) of the calibration curve were 0.997 and 0.999 for DON and ZEA, respectively. Spiking was performed in triplicates at three levels in the barley matrix. In addition, the spiking experiments for the *F. graminearum* strains grown in culture media were carried out at two levels.

The method is suitable for the determination of mycotoxins in barley as the given LOQ’s are lower than the maximum limit set by the Brazilian and EU regulations for the content of DON and ZEA. The parameter linearity, reproducibility, repeatability and recovery obtained were also shown to be adequate.

### 2.3. Mycotoxins Analysis

The analysis carried out in the present research revealed the occurrence of the two mycotoxins most commonly found in barley. Grain samples that presented levels above the LOQ were considered positive. Mean calculations were performed using Microsoft^®^ Excel 2007 only including positive samples. Deoxynivalenol showed the highest incidence (90.6%) with levels ranging from 45.95 to 1155.21 µg/kg. ZEA also had a high occurrence (87.5%) with values ranging from 82.41 to 423.71 µg/kg. Regarding the Brazilian regulation for DON, only one sample was above the established maximum levels (1000 µg/kg). Nevertheless, for ZEA, 55 samples (86%) were above the regulation (100 µg/kg) (Table 2).

Additionally, the mycotoxin production from toxigenic potential *F. graminearum* strains was observed. The frequency of the DON strain producers was 80% (12 strains) and only 20% (3 strains) for ZEA. The levels ranged from 123.03 to 592.61 µg/kg and 33.64 to 140.58 µg/kg, respectively.

## 3. Discussion

This study has shown that the majority of the *Fusarium* species isolated from brewing barley grains belonged to the FSAMSC, whereas the main trichothecene producing species are clustered. Mycotoxins analysis demonstrated that most of the samples were contaminated with DON (90.6%) and ZEA (87.5%), highlighting the importance of this investigation.

*Fusarium* species are found in cereal grains, such as barley, wheat, maize and rice worldwide, where mycotoxins can be found in high concentrations [15,16,17,18]. This fact, may be worsened by weather conditions, such as high humidity and temperatures that tend to increase *Fusarium* infection in plants [19,20]. Humidity is an important environmental factor and it influences the water activity of the grains. This intrinsic factor is important for fungal growth and has considerable association with mycotoxin production [21].

Generally, *Fusarium* development as well as DON and ZEA production can be seen with higher levels of water activity (0.90) [22]. Nevertheless, the current research showed a low variation of the water activity that was observed among the 64 brewing barley samples, with mean value of 0.622 ± 0.02. In this case, both the germination of fungal spores and the growth of storage fungi are inhibited. However, the grain analyses were carried out after the cleaning and drying stages, explaining the levels found. For safe storage of grain, the grain moisture content must be compatible with the period of time the grain will be stored in order to avoid the fungi growth and therefore, the water activity should be less. To associate *Fusarium* mycotoxins found in this study it is necessary to assume that the DON and ZEA production could be correlated to the fungi presence and high water activity in the growing plant stages. This fact could be explained in our recent study with rice grains, where the levels of ZEA found are associated with the presence of *Fusarium* during pre-harvest, in grains freshly harvested with high levels of moisture content and water activity. After the food processing steps were completed in the industry, there was not any *Fusarium* growth in the grains, however, the ZEA levels remained in the parboiled rice (water activity: 0.64 ± 0.02), resisting the degradation [23].

The phylogenetic study showed that the majority of the strains clustered with *F. graminearum*, *F. poae* and *F. avenaceum*. *Fusarium poae* can produce high levels of nivalenol; therefore, further analyses should be done to investigate the degree of nivalenol contamination in Brazilian barley. Previous studies carried out by [24] in Russia and [25] in Italy have shown a high incidence of *F. avenaceum* in barley as well as other *Fusarium* species. [26] observed a high incidence of *F. avenaceum*, *F. graminearum* and *F. culmorum* in in Finnish barley grains between the years of 2005–14, with high levels of trichothecene contamination. The presence of *F. avenaceum* in barley may indicate the presence of enniatins, moniliformin and beauvericin in the samples. In Brazil, another study in barley has demonstrated that the majority of the isolates belonged to the *F. graminearum* lineage; however, mycotoxin analysis was not performed in that study [27]. To our knowledge, this is the first report of phylogenetic identification of the *Fusarium* species in Brazilian barley and the first correlation with DON and ZEA contamination.

With respect to DON contamination, the current study showed similarities with those found in a survey carried out in Spain with a mean level of 119.9 µg/kg [28]. In addition, the highest results were obtained by [25] reporting incidence and a maximum contamination level of 108.7 µg/kg. Also, [29] in Italy, a study reported lower incidence and concentrations of DON with a maximum level of 35.5 µg/kg; and [30] in Tunisia evaluated its presence with a maximum level of 6.1 µg/kg. In the current study, only one sample presented a high level of DON (1155.21 µg/kg) and was above the established maximum level set by the Brazilian regulation. The samples showed mean values of 147.65 µg/kg and median values of 98.68 µg/kg that also demonstrated low levels.

In contrast, the ZEA contamination found in the present study were of significance, considering the maximum levels established by the Brazilian and international regulation, where 86% of the samples were above [13,31]. The samples showed mean values of 123.24 µg/kg and median values of 119.26 µg/kg. In a study performed by [32] in the Czech Republic, a few samples were contaminated with ZEA with values ranging from 181.2 to 204.4 µg/kg, which is quite similar to the results of our study. Furthermore, high levels (max 985.9 µg/kg) of ZEA were found in another survey carried out in the Czech Republic with samples from the 2011 crop [33].

Two other studies were conducted in Brazil by the current author and can be compared to this study. The first one was from the 2014 crop and low levels of DON were found ranging from 200 µg/kg to 15.000 µg/kg [34]. The other one was from the 2015 crop and higher levels were found for both toxins, DON and ZEA. The last study was considered an issue for the industry due to the levels exceeding the regulation levels established. The mean levels ranged from 1700 to 7500 µg/kg and from 300 to 630 µg/kg for DON and ZEA, respectively.

It is necessary to mention that not only the large-scale brewing industry is increasing but that the craft breweries in Brazil are expanding. These small brewing groups always look for the best sources and have the characteristic of “German beer purity law,” meaning that just barley is used for beer production. For these reasons, barley needs to be of higher quality. On the other hand, the large-scale industry in Brazil uses other grains such as corn, rice and sorghum for beer production which are considered low quality grains [35]. Some studies were carried out and they showed contamination with fumonisin B_1_ [36] that is commonly found in corn and its derivatives [37].

In the last years the stability of these metabolites has been studied. Deoxynivalenol showed to be a mycotoxin that persists through the process and demonstrated stability in some industry processes, such as cleaning, milling, brewing and extrusion [9]. Zearalenone has had some studies published about it which showed lower stability. However, its levels are high when they are found. A research conducted by [38] showed a considerable reduction of ZEA levels in the presence of *Saccharomyces cerevisiae* yeast. The main point is even if there is a significant reduction of these metabolites, sometimes they still offer a risk, especially when in beer, which is considered one of the most consumed beverages in the world.

Another aspect that should be taken into account is the masked toxins. ZEN-14-sulfate and DON-3-glucoside are most commonly observed in grains. They could be present in the matrix, however masked toxins are either bound to carbohydrates or proteins and, therefore, are not extractable with existing protocols aimed at the extraction of the toxin, or they are not detectable using established chromatography routines; hence their name “masked” mycotoxins [39]. Further studies are being planned to gain more knowledge on these metabolites in barley and also in beer.

The *Fusarium* mycotoxins found in barley grains and the toxigenic potential analysis of the *F. graminearum* strains isolated in this study reinforce the importance of these genera in this relevant commodity. Furthermore, the identification of *F. poae*, *F. avenaceum* and the genus *Alternaria* in barley samples highlights the importance of further research on other mycotoxins in barley and its by-products. *Alternaria* species were recovered from 28% of the barley samples. This may have important implications on other mycotoxins that may be found in high concentrations, such as tenuazonic acid. Further studies should be conducted in order to evaluate the co-occurrence of the mycotoxins produced by this genus [25,40].

In this study, the toxigenic potential of the *F. graminearum* strains isolated in the brewing barley was evaluated and both toxins detected in the samples were produced by them. In total, 80% and 20% of the strains produced DON (mean: 297.02 µg/kg and median: 268.74 µg/kg) and ZEA (mean: 79.7 µg/kg and median: 64.86 µg/kg), respectively. The toxins levels found were lower when compared to that of a study performed by Wu et al. (2017), in which the strains exhibited a production of 1405.05 µg/kg for DON and 4118.31 µg/kg for ZEA. The parameter temperature is crucial for DON production and some studies showed a variation of the optimal value, that varies between 20 and 28 °C [19,20,41]. On the other hand, temperature negatively affected ZEA production. The optimized condition for ZEA production was cultivation at 15 °C [41]. The toxin combination can be related to several mycotoxigenic fungi that contaminate barley in the field. However, the mycotoxin occurrence in Brazilian barley suggests high prevalence of toxigenic *F. graminearum* and related species, which could explain the DON and ZEA levels in almost 85% of the samples.

## 4. Conclusions

The DON and ZEA contamination in brewing barley grains were detected in 90.6% and 87.5% of the samples. The phylogenetic study showed that the majority of the strains clustered with *F. graminearum*, *F. poae* and *F. avenaceum*. Toxigenic species of *F. graminareum* isolates presented a higher percentage in the samples (31.25%) and can explain the DON and ZEA contamination found in the barley samples. Taking into account the Brazilian regulation of ZEA levels in barley, 86% of the samples were significantly higher than the current maximum limit, while for DON, only one sample was above the established maximum levels. The Brazilian regulation for mycotoxins will be updated by 2019 and therefore, the new maximum limits are still under discussion, based on the analysis of the largest amount of data available of the occurrence of mycotoxins in the grains produced in Brazil. This monitoring data of the toxigenic *Fusarium* and its mycotoxins could lead to greater knowledge of the current situation of the barley contamination in the industry, which can assist Brazilian regulation and the programing of management strategies in order to avoid the toxic effects on human and animal health.

Fungal infection and the presence of mycotoxins in cereals is natural and the prevention of these occurrences is difficult even if good agricultural practices are maintained. The data provided in this study was important for the knowledge on *Fusarium* diversity and toxin contamination. Furthermore, the results highlight the importance of monitoring DON and ZEA contamination in barley grains during pre, post-harvest and in processed food, such as beer, mainly for the development of management strategies. Consequently, serious economic losses and health problems could potentially be avoided.

## 5. Materials and Methods

### 5.1. Barley Samples

A total of 64 brewing barley (BRS Brau variety) samples were obtained from the 2016 harvest, from the States of Paraná and Rio Grande do Sul, the largest barley-producing regions in Brazil. Samples were collected from bulk batches, after dirt removal and drying (up to 60 °C) in the storage units. Sampling was performed using a grain auger from different points of the bulk batches, with a minimum final weight of 5 kg. Each sample was homogenized, reduced into portions of 1.0 kg to be representative of the overall sample and further was milled for each analysis. Samples were packed in polyethylene bags and stored at 4 °C and different amounts were used for mycobiota and mycotoxin analyses.

### 5.2. Water Activity

To perform water activity (aw) analysis, 2 g of each barley sample were submitted to Aqua-Lab 4TE equipment Aqua-Lab 4TE, Decagon Devices (Sao Jose dos Campos, SP, Brazil). Samples were analyzed in triplicate according to the Association of Official Analytical Chemists—[42].

### 5.3. Mycobiota and Identification of Fungi

The dilution technique was used for fungal isolation, as described by [43]. To summarize briefly, twenty-five grams of each sample were added to 225 mL of 0.1% peptone dissolved in water in sterile conditions. The mixture was stirred on a rotary shaker for 2 min., dilutions of 10^−1^, 10^−2^, 10^−3^ and 10^−4^ were obtained, 0.1 mL aliquots of each dilution were spread on the PDA medium [9,10] containing chloramphenicol (100 mg/L) (in duplicate). These were incubated for 5 days, at 25 °C in the dark. The results were expressed into colony forming units per gram (CFU/g) in the dilution 10^−1^, as the colonies were easily distinguished in this dilution factor. The isolates were identified morphologically according to Pitt and Hocking, (2009).

### 5.4. Identification of the Fusarium Species

The strains were grown in yeast extract sucrose (YES) agar [44] for 3 days at 25 °C. The DNA was extracted using DNeasy Plant Mini Kit (Qiagen, Hilden, Germany) according to the manufacturer’s instructions.

The partial sequences of elongation factor (*EF-1α*) and the second fragment of *RPB2* (*7CF/11AR*) were selected in order to identify the *Fusarium* isolates. The amplification of the *EF-1α* and *RPB2* loci were performed according to [45,46]. Amplicons were purified with ExoSAP-IT (Affymetrix, Santa Clara, CA, USA) and sent to the Centre of Human Genome Studies, University of Sao Paulo, Brazil for sequencing in ABI PRISM 3130 DNA Analyzer (Applied Biosystems, Foster City, CA, USA).

Sequences were aligned using the multiple alignment software ClustalX v. 1.83 plug-in in the software Geneious v. 5.3.6 (Biomatters, Auckland, New Zealand). The alignments were edited using the sequence alignment-editing program Geneious v. 1.83 and each polymorphism was re-examined by checking the chromatograms. The sequences generated in this study were deposited in the GenBank (Table 1 and Appendix A).

### 5.5. Phylogenetic Analysis

Phylogenetic analysis was performed based on the *EF-1*α and *RPB2* combined datasets using the PAUP 4.0b10 (Sinauer Associates, Sunderland, MA, USA) [47]. Phylogenies were obtained by using Unweighted Parsimony analysis and heuristic search option with 1000 random addition sequences and tree bisection reconnection branch swapping in PAUP 4.0b10 [48]. Gaps were treated as missing data. The Consistency Index (CI) and the Retention Index (RI) were calculated to indicate the amount of homoplasy present. Clade stability was assessed via bootstrap analysis in PAUP 4.0b10, using 1000 heuristic search replications with random sequence addition. The data sets were rooted with *Fusarium* sp. as it is considered a suitable out-group [49]. The reference sequences for the *Fusarium* species used in this study were obtained from NCBI (Appendix A).

### 5.6. Mycotoxin Analysis

#### 5.6.1. Chemicals and Reagents

Both standards (DON and ZEA) were purchased from Sigma Aldrich Chemicals (St. Louis, MO, USA). Stock solution standards were prepared in methanol at concentrations of 1 mg/mL for DON and ZEA. From the individual stock standard solutions, a standard mixture was prepared at the following concentrations: 0.025, 0.0375, 0.0625, 0.125, 0.375, 0.500 µg/mL. The standard mixture was prepared in methanol and stored at −18 °C. Methanol and acetonitrile (LC-MS/MS grade) were supplied by J.T Baker (Sao Paulo, SP, Brazil). Acetic acid was obtained from Biotec (Pinhais, PR, Brazil). High-purity Milli-Q water (18.2 MΩ/cm) was obtained from the Millipore Synergy system (Billerica, MA, USA).

#### 5.6.2. DON and ZEA Extraction

Mycotoxin extraction was carried out according to [50], with some minor modifications. Briefly, 2 g of brewing barley were ground and homogenized in 8 mL of acetonitrile:water (80:20 *v*/*v*) and shaken for 60 min. The mixture was then centrifuged for 10 min at 3500 rpm. The supernatant was transferred to an amber vessel and dried using a heating block and a nitrogen stream. The dried extract was resuspended in 500 µL of a mobile phase consisted of 70% of water:methanol:acetic acid (94:5:1, *v*/*v*/*v*) and 30% of water:methanol:acetic acid (2:97:1, *v*/*v*/*v*). Finally, 5 µL was injected in the LC-MS/MS system for analysis.

### 5.7. Production of DON and ZEA by the Strains

The isolates of *F. graminearum* were grown onto PDA (three agar plugs, 6 mm in diameter) and tested for DON and ZEA production. The culture media was incubated at 24 °C and 15 °C with a moisture content of 90% and 80% for 20 days for DON and ZEA, respectively [51,52]. The mycelium was transferred into an Erlenmeyer flask containing 30 mL of chloroform and shaken for 60 min for mycotoxin extraction, followed by filtration through anhydrous sodium sulfate (Na_2_SO_4_). The extract was dried and re-suspended with 500 µL of mobile phase. Finally, the extract was filtered with a syringe filter (nylon membrane 0.22 µM). The sample was quantified by liquid chromatography/mass spectrometry (LC-MS/MS) for DON/ZEA.

### 5.8. Chromatography Conditions

Detection and quantification were carried out using an LC-MS system from Thermo Scientific^®^ (Bremen, Germany) composed of an ACCELA 600 quaternary pump, an ACCELAAS auto-sampler and a triple quadrupole mass spectrometer TSQ Quantum Max.

The chromatographic conditions were performed according to [53]. In short, the following instrumental settings were applied: the triple quadrupole mass spectrometer TSQ Quantum Max was operated at positive polarity and the ionization conditions were 208 °C for capillary temperature, 338 °C for vaporizer temperature, 4500 V for spray voltage and 60 arbitrary units for sheath gas pressure. For selectivity, the mass spectrometer was operated at MRM mode monitoring, three transitions per analyte, using a collision gas pressure of 1.7 mTorr and collision energy (CE) ranging from 11 to 40 eV.

The mass spectrometric conditions were optimized by re-tuning different analytes by direct infusion of each analyte individually. The tube lens potential, collision energies and product ions were optimized and carefully chosen. The most abundant mass-to-charge ratio (*m*/*z*) was selected for each compound of interest. The mycotoxins exhibited precursor ions and product ions with reasonably high signal intensities in positive ESI mode (ESI+) and protonated molecules [M + H] were found. Appendix A shows the retention times (tR), MRM transitions as well as the tube lens potential and collision energies optimized for each compound.

Separation was performed on a C8 Luna column, with a particle size of 3 µm, 150 × 2.0 mm, length and diameter, respectively, Phenomenex (Torrance, CA, USA). In the mobile phase, solvent A (water:methanol:acetic acid, 94:5:1, *v*/*v*/*v*) and solvent B (water:methanol:acetic acid, 2:97:1, *v*/*v*/*v*) were used. The gradient program was applied at a flow rate of 0.2 mL/min under the following conditions: 0–1 min 55% B; 1–3 min 55–100% B; 3.01–7 min 100% B and 7.01–12 min 55% B. The total analytical run time was 7.5 min for the 2 toxins (Table 3).

#### Method Validation

The methods for extraction of mycotoxins in brewing barley and in culture media with *Fusarium* growth were validated according to the Commission Regulation [54] guideline. To determine the limit of detection (LOD), limit of quantification (LOQ), recovery, repeatability and selectivity/specificity, samples with non-detectable levels of mycotoxins were submitted to spiking experiments.

Considering linearity, a six-point calibration curve was constructed with the following concentrations of the mycotoxin standard mixture (DON and ZEN): 0.025, 0.0375, 0.0625, 0.125, 0.375, 0.500 µg/mL. The LOD and LOQ methods were determined by fortifying blank samples with different concentration levels and the experiments were repeated on three different days. The LOD was defined as the minimum concentration of an analyte in the spiked sample with a signal noise ratio equal to 3 and LOQ with a signal noise ratio equal to 10.

### 5.9. Data Analysis

Results regarding DON and ZEA in brewing barley samples and *F. graminearum* strains were reported as the mean ± standard deviation and median, using Microsoft office Excel 2007.

## Figures and Tables

**Figure 1 toxins-11-00031-f001:**
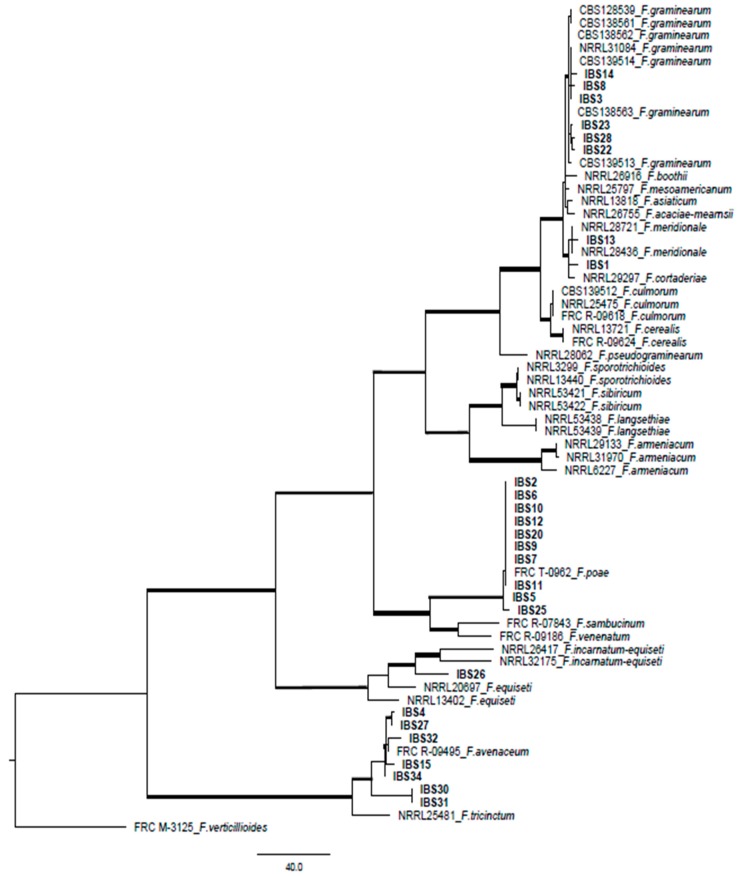
LC-MS/MS method performance.

**Table 1 toxins-11-00031-t001:** Frequency of the members of each *Fusarium* species complex isolated from brewing barley grains.

Species Complex *	Frequency %	*Fusarium* Species	% Samples Contaminated
FSAMSC	56.25	*F. graminearum*	23.4
		*F. poae*	17.2
		*F. meridionale*	1.6
FTSC	31.25	*F. avenaceum*	23.4
FFSC	8.33	*F. proliferatum*	3.1
		*F. verticillioides*	3.1
FIESC	2	*F. incarnatum-equiseti*	1.6
FOSC	2	*F. oxysporum*	1.6

* FSAMSC: *F. sambucinum* species complex, FTSC: *F. tricinctum* species complex, FFSC: *F. fujikuroi* species complex, FIESC: *F. incarnatum-equiseti* species complex, FOSC: *F. oxyxporum* species complex.

**Table 2 toxins-11-00031-t002:** DON and ZEA contamination in barley grains.

	Number Samples	Deoxynivalenol				Zearalenone			
		Positive Samples/% *	Range of Positive Samples (µg/kg)	Mean ± SD (µg/kg)	Median (µg/kg)	Positive Samples/% *	Range of Positive Samples (µg/kg)	Mean ± SD (µg/kg)	Median (µg/kg)
Barley grains	64	58/90.6	45.95–1155.21	147.65 ± 167.16	98.68	56/87.5	82.41–423.71	123.24 ± 45.29	119.26

* > LOQ of 25 µg/kg.

**Table 3 toxins-11-00031-t003:** Retention time and mass spectrometric parameters used in the analysis of the mycotoxins.

Mycotoxin	Retention Time (min)	Precursor ion (*m*/*z*)	Product ion (*m*/*z*) *	CE (V)	TubeLens
DON	2.19	297 [M + H]	203Q	17	71
			175C	18	71
			91C	39	71
ZEA	6.55	319 [M + H]	283Q	11	79
			187C	25	79
			185C	20	79

* Q, Quantification transition C, Confirmation transition.

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
