# Peer review of "Assessment of Toxigenic Fusarium Species and Their Mycotoxins in Brewing Barley Grains"

_toxins, 2019, doi:10.3390/toxins11010031_

Round 1

Reviewer 1 Report

The data presented in the current manuscript are useful to monitor mycotoxins in the grains subsequently used for brewing. The methods used are appropriate and the data presented support the conclusions. Please find a few minor corrections below.

Line 117, 127: Please do not use citation in the results section

Line 124, 200: Please do not start a sentence with abbreviation; Spell out ‘DON’

Line 200: Spell out ZEA

Line 222: Add ‘the’ after ‘In’

Line 275: check spacing

Author Response

REVIEWERS' COMMENTS:

Dear Reviewers, 

We had performed the necessary adjustments and they certainly improved the quality of our manuscript. All of the changes are highlighted in red in the text. We hope that the text below answers your questions satisfactorily.Thank you very much for your suggestions.

REVIEWER #1

The data presented in the current manuscript are useful to monitor mycotoxins in the grains subsequently used for brewing. The methods used are appropriate and the data presented supports the conclusions. Please find a few minor corrections below.

Line 117, 127: Please do not use citation in the results section

Line 124, 200: Please do not start a sentence with abbreviation; Spell out ‘DON’

Line 200: Spell out ZEA

Line 222: Add ‘the’ after ‘In’

Line 275: check spacing

Answer: All the corrections were done. 

Reviewer 2 Report

I believe the manuscript needs some minor modifications before publishing. At some places the choice of words can be improved. F.i., on Line 15 "Nevertheless," does not really make sense because what follows is not contrasting the previous statement but adding to it.

Line 34:  in the context of animal nutrition one speaks of feed, while food is associated to human nutrition.

L36: mycotoxin with estrogenic effects would be better than "estrogenic mycotoxin".

L45: It is not the industry but the beer gaining increased popularity.

L60: Here the "however" does not make sense.

L65: duly instead of "dully".

L164: insert "in" before "Finnish".

In the description of the MS conditions "bar" is used on L336. The gas flows for Thermo MSs are set in arbitrary units and not bar. On L 345 you use the term "cone voltage" which is a Waters term. Thermo MS do not have a cone voltage but a tube lens potential. On L 352 the analytical run time is stated as 7.5 min while the gradient time is 12 min. Table S2 should appear in the main body of the manuscript since it is of immediate importance to the reader.

The reference 14 "Commision Regulation (EC) No 401/2006" does neither provide any legislative limit, as implied on L 78, nor does it provide guidance for method validation, as implied on L 355.

On L 265 equipment (Aqua-Lab 4TE) is mentioned without providing source information.

On L 294 "PAUP 4.0" is mentioned the first time while providing source information two lines further down.

Author Response

REVIEWERS' COMMENTS:

Dear Reviewers, 

We had performed the necessary adjustments and they certainly improved the quality of our manuscript. All of the changes are highlighted in red in the text. We hope that the text below answers your questions satisfactorily.Thank you very much for your suggestions.

REVIEWER #2

I believe the manuscript needs some minor modifications before publishing. In some instances the word choicecan be improved. F.i., on Line 15 "Nevertheless," does not really make sense because what follows is not contrasting the previous statement but instead adding to it. 

Line 34:  in the context of animal nutrition one speaks of feed, while food is associated withhuman nutrition.

Answer:The sentence was corrected 

L36: mycotoxin with estrogenic effects would be better than "estrogenic mycotoxin".

Answer: Corrected 

L45: It is not the industry but the beer gaining the increased popularity.

Answer: Corrected 

L60: Here the "however" does not make sense.

Answer: Withdraw 

L65: duly instead of "dully".

Answer: Corrected 

L164: insert "in" before "Finnish".

Answer: Added

In the description of the MS conditions "bar" is used on L336. The gas flows for Thermo MSs are set in arbitrary units and not bar. On L 345 you use the term "cone voltage" which is a Waters term. Thermo MS do not have a cone voltage but a tube lens potential. On L 352 the analytical run time is stated as 7.5 min while the gradient time is 12 min. Table S2 should appear in the main body of the manuscript since it is of immediate importance to the reader.

Answer: Corrected

The reference 14 "Commision Regulation (EC) No 401/2006" does neither provide any legislative limit, as implied on L 78, nor does it provide guidance for method validation, as implied on L 355.

Answer: The references were verified

On L 265 equipment (Aqua-Lab 4TE) is mentioned without providing source information.

Answer: Added

On L 294 "PAUP 4.0" is mentioned the first time while providing source information two lines further down.

Answer: Corrected

Reviewer 3 Report

In this work, 64 malting barley samples from Brazil were analysed for their DON and ZEN content, as well as for their Fusarium incidence. The incidence was high for both mycotoxins, while the mean concentration of ZEN was closer to the maximum permitted levels than that for DON.

A previous article exists which reports on similar results:

Piacentini, K.C., Rocha, L.O., Savi, G.D., Carnielli-Queiroz, L., Almeida, F.G., Minella, E., Corrêa, B.

Occurrence of deoxynivalenol and zearalenone in brewing barley grains from Brazil

(2018) Mycotoxin Research, 34 (3), pp. 173-178.

Thus there is not much novelty in the work. Another problem is that it is not possible to assess from the work description the amount of work carried out and its representability. The results could be summarised in a short communication.

Line 262, which amount of sample was ground? 1 kg portions? What was the aim of keeping 1kg portions?

Section5.3., using this methodology surface contaminating fungi (not infecting) are determined, which may not be relevant. Perhaps different distributions of contaminating species would have been obtained if previous surface disinfection had been carried out.

Section 5.4, how many isolates were tested? All of them? If not, how were they selected?

Line 321, from my experience PDA does not favour, in general, mycotoxin production

Lines 87-88, in my opinion, % contaminated samples is a more interesting data than % of isolates belonging to the different genera.

Table 1, please add as a footnote the species complex in full. Also include the % samples contaminated by each species

In table 2, the last row is not clear, does it refer to toxin production by isolates of fFusarium graminearum? What do these numbers (15, 12/80, 3/20) mean? I don’t think it is a good idea to include these data under the same headings

Finally, It is required to add results of Fusarium presence in the different samples, and analysis of correlation between Fusarium species and DON or ZEN concentration must be shown. As samples were not stored after harvest for a longtime, a certain correlation should exist.

Author Response

REVIEWERS' COMMENTS:

Dear Reviewers, 

We had performed the necessary adjustments and they certainly improved the quality of our manuscript. All of the changes are highlighted in red in the text. We hope that the text below answers your questions satisfactorily.Thank you very much for your suggestions.

REVIEWER #3

In this work, 64 malting barley samples from Brazil were analysed for their DON and ZEN content, as well as for their Fusarium incidence. The incidence was high for both mycotoxins, while the mean concentration of ZEN was closer to the maximum permitted levels than that for DON. A previous article exists which reports on similar results: Piacentini, K.C., Rocha, L.O., Savi, G.D., Carnielli-Queiroz, L., Almeida, F.G., Minella, E., Corrêa, B. Occurrence of deoxynivalenol and zearalenone in brewing barley grains from Brazil. (2018) Mycotoxin Research, 34 (3), pp. 173-178. 

Thus there is not much novelty in the work. Another problem is that it is not possible to assess from the work description the amount of work carried out and its representability. The results could be summarized in a short passage.

Answer: The current study can be considered innovative with relation to the previously published work. This is due to the first identification of phylogenetic identified Fusariumspecies contamination in Brazilian barley with an evidence of a possible association of DON and ZEA present in the samples. Due to ZEA levels being significantly higher than the current maximum limit in Brazilian regulation, it is important to know the fungi species in order to find effective methods for their control in barley, especially when growing plant stages can be highly contaminated with the mycotoxin production. This represents an actual relevance to the country due to concern about the maximum levels of mycotoxins which will be updated 2019. As well as import and export barley products, including brewing barley.

Line 262, which amount of sample was ground? 1 kg portions? What was the aim of keeping 1kg portions?

Answer: Some information was added. However, for each analysis (mycobiota and mycotoxins) a different amount of ground grains was used. Portions of 1 kg were used to be representative of the overall sample. Considering that, after milling, they are again homogenized and weighed for each type of analysis.

Section 5.3., using this methodology surface contaminating fungi (not infecting) are determined, which may not be relevant. Perhaps different distributions of contaminating species would have been obtained if previous surface disinfection had been carried out.

Answer:When we first started the experiments, we tried both methodologies (surface contaminating and disinfection). We compared the methodologies and we recovered more isolates by using the dilution technique. We did not mention in Materials and Methods section that milled samples where used, in order to obtain a larger contact surface between the grain and peptone water. Therefore, more isolates of fungi were recovered. The word “milled” was added into materials and methods section, line 273. In addition, most part of the isolates found in the samples were from toxigenic fungal species that in the present study can be possibly related with the mycotoxin accumulation found in barley.

Section 5.4, How many isolates were tested? If not all of them, how were they selected?

Answer:The identification analysis was carried out with 48 Fusariumisolates. We isolated approximately 90 Fusariumstrains. The 48 isolates were chosen randomly, because our laboratory had funding to do 200 sequencing reactions, therefore 50 isolates at most (considering EF1áand RPB2 loci).

Line 321, from my experience PDA does not favour, in general, mycotoxin production

Answer:We followed the protocol described by Pagnussatt et al. (2014), Inhibition of Fusarium graminearum growth and mycotoxin production by phenolic extract from Spirulina sp., Pesticide Biochemistry and Physiology, p. 21-26; and Heidtmann-Bemvenuti et al. (2015), Effect of natural compounds on Fusarium graminearum complex, Science of Food and Agriculture. We also tested this medium before starting the experiments. Perhaps PDA is not the best medium for mycotoxin production, but we could recover DON and ZEA consistently. In addition, according to previously published studies, it is possible to confirm that PDA is a suitable medium for the production of toxins, including from toxigenic Fusariumspecies (Savi et al. 2013a;b).

Savi, G.D.; Vitorino, V.; Bortoluzzi, A.J.; Scussel, V.M. Effect of zinc compounds on Fusarium verticillioides growth, hyphae alterations, conidia, and fumonisin production. J. Sci. Food Agric. 201393, 3395–3402, doi:10.1002/jsfa.6271.

Savi, G.D.; Bortoluzzi, A.J.; Scussel, V.M. Antifungal properties of Zinc-compounds against toxigenic fungi and mycotoxin. Int. J. Food Sci. Technol.201348, 1834–1840, doi:10.1111/ijfs.12158.

Lines 87-88, in my opinion, the percent ofcontaminated samples provides more useful data than the percentof the different genera isolates.

Answer:The focus of the current study was the isolation of Fusariumgenus. However, we would like to add this data about other genera isolated only for reader knowledge considering general mycobiota of barley grains.

Table 1, please add as a footnote the species complex in full. Also include the percent ofsamples contaminated by each species

Answer:The information was added. 

In table 2, the last row is not clear, does it refer to toxin production by isolates of Fusarium graminearum? What do these numbers (15, 12/80, 3/20) mean? I don’t think it is a good idea to include thisdata under the same headings

Answer:The data about toxins production was added in the text. The Table 2 was reorganized.

Finally, It is required to add results of Fusarium presence in the different samples, and an analysis of correlation between Fusarium species and DON or ZEN concentration must be shown. As samples were not stored after harvest for a long time, a certain correlation should exist. 

Answer:The majority of the isolated strains belonged to the FSAMSC, and within this species complex, we showed within the work that the isolated F. graminearum was able to produce DON and ZEA. Only 24% of the isolates were identified as F. graminearum, therefore we do not think that a correlation test would bring another significative result. If we had more F. graminearum isolates, this test would provideusefulresults.

Round 2

Reviewer 3 Report

All comments were adressed in the answers, however, the informations should be tranferred to the manuscript. For example, the number of strains involved in the work, I could not find it in the revised manuscript, and I feel it's unfair for the reader, as it is required to assess the representativity of the study.

Author Response

Reviewer 3#

All comments were adressed in the answers, however, the informations should be transferred to the manuscript. For example, the number of strains involved in the work, I could not find it in the revised manuscript, and I feel it's unfair for the reader, as it is required to assess the representativity of the study.

Answer:The information that we answered in the last review was added to the text (highlighted in yellow) as recommended.

Thank you for the comments, certainly they improved the manuscript.  
